# Role of RUNX3 in Restriction Point Regulation

**DOI:** 10.3390/cells12050708

**Published:** 2023-02-23

**Authors:** Jung-Won Lee, You-Soub Lee, Min-Kyu Kim, Xin-Zi Chi, Dohun Kim, Suk-Chul Bae

**Affiliations:** 1Department of Biochemistry, College of Medicine and Institute for Tumour Research, Chungbuk National University, Cheongju 28644, Republic of Korea; 2Department of Thoracic and Cardiovascular Surgery, College of Medicine, Chungbuk National University, Cheongju 28644, Republic of Korea

**Keywords:** RUNX3, K-RAS, R-point, cell cycle

## Abstract

A cell cycle is a series of events that takes place in a cell as it grows and divides. At the G_1_ phase of cell cycle, cells monitor their cumulative exposure to specific signals and make the critical decision to pass through the restriction (R)-point. The R-point decision-making machinery is fundamental to normal differentiation, apoptosis, and G_1_–S transition. Deregulation of this machinery is markedly associated with tumorigenesis. Therefore, identification of the molecular mechanisms that govern the R-point decision is one of the fundamental issues in tumor biology. *RUNX3* is one of the genes frequently inactivated in tumors by epigenetic alterations. In particular, *RUNX3* is downregulated in most *K-RAS*-activated human and mouse lung adenocarcinomas (ADCs). Targeted inactivation of *Runx3* in the mouse lung induces adenomas (ADs), and markedly shortens the latency of ADC formation induced by oncogenic *K-Ras*. RUNX3 participates in the transient formation of R-point-associated activator (RPA-RX3-AC) complexes, which measure the duration of RAS signals and thereby protect cells against oncogenic RAS. This review focuses on the molecular mechanism by which the R-point participates in oncogenic surveillance.

## 1. The Restriction Point

The major events that regulate cell proliferation occur during the G_1_ phase of the cell cycle. The growth of normal cells in culture is regulated by complex interactions involving growth factors, cell density, and cell attachments to substrates. Growth factors are necessary to initiate and maintain the transition through the G_1_ phase, leading to the S phase. Reductions in growth factor levels in cells, such as by the removal of serum, prevent the onset of the S phase [1]. However, once the cells have moved through a certain G_1_ decision-making period, the removal of serum no longer affects their progress through the cell cycle, with these cells proceeding through the remainder of the G_1_ phase and onward through S, G_2_, and M phases. The point in G_1_ at which commitment occurs and the cell no longer requires growth factors to complete the cell cycle has been termed the restriction (R)-point [1]. Once beyond the R-point, cells are committed to DNA synthesis and are independent of extracellular growth factors during the remainder of the cell cycle (Figure 1). R-point transition is regulated by R-point-associated proteins (R-proteins), including c-Myc, cyclins, CDKs, p21, p27, E2F, and pRB [2], with pRB serving as the primary molecular regulator [3].

One of the most important breakthroughs in the understanding of cell cycle regulation was the finding that mitogenic stimulation was connected to the cell cycle machinery. The expression of cyclin D and its assembly with CDK4 and CDK6 into active kinase complexes are regulated by growth factors [4,5], indicating that cyclin D is a growth factor sensor. In turn, the ability of cyclin D-dependent kinases to trigger the phosphorylation of pRB during the mid- to late-G_1_ phase of the cell cycle makes inactivation of the growth-suppressive function of pRB a mitogen-dependent step. pRB participates in controlling the G_1_/S-phase transition. Cyclin E complexes with CDK2 downstream of the cyclin D-CDK4/6 complex, with the cyclin E-CDK2 complex further phosphorylating pRB. This shift from cyclin D-CDK4/6 to cyclin E-CDK2 accounts for the loss of dependency on growth factors, indicating that the R-point lies between cyclin D-CDK4/6 and cyclin E-CDK2 (Figure 1).

The mammalian G_1_/S cell cycle phase transition network is a highly nonlinear network that produces seemingly paradoxical results [6]. Numerous feedback loops lead to situations in which downstream events lie upstream of themselves. For example, pRB2/p130 and p27 are both involved in a negative feedback regulatory loop with cyclin E [7]. Moreover, although *c-Myc* expression is downstream from p21, CDKs, and E2F [8,9], *c-Myc* is also an upstream regulator of *p21* and *CDKs* [10]. Similarly, the finding that *cyclin E* is transactivated by E2F-1 suggests that *cyclin E* is located downstream from pRB and E2F-1. However, cyclin E inactivates pRB and releases E2F. These positive loops ensure the irreversibility of commitments. Once expressed, cyclin E becomes independent of downstream growth-factor-dependent cyclin D1. The phosphorylation of pRB abrogates growth factor dependency, enabling the cells to pass through the R-point and commit to completing the remaining phases of the growth cycle [11]. The nonlinear networks include c-Myc, cyclins, p21, CDKs and E2F, which play key roles in R-point regulation. Therefore, the nonlinear networks producing seemingly paradoxical results appears to be associated with R-point, which regulates cell fate.

## 2. Regulation of the Timing of the R-Point

The R-point lies between cyclin D–CDK4/6 and cyclin E–CDK2, suggesting that cyclin E–CDK2 is responsible for pushing the cells from the R-point through to the remainder of the G1 phase. Therefore, cyclin E–CDK2 complexes should form only after the cell is committed to proliferation at the R-point. Although *cyclin D* expression is induced earlier than *cyclin E* expression, the times of their expression were found to overlap considerably [3] (Figure 1), suggesting that the expression of *cyclin E* does not solely determine the time of exit from the R-point. Cyclin E–CDK2, however, is activated only after R-point commitment to proliferation. p21^WAF1/CIP1/Kip^ (hereafter p21), originally identified as an inhibitor of CDKs [12], is encoded by an immediate-early gene, with p21 mRNA peaking approximately 2 h after stimulation with serum or growth factor [13]. However, the biological importance of the mitogen-stimulated immediate early induction of *p21* was not understood at that time. Subsequent studies revealed that p21 is not a simple CDK inhibitor; rather, members of the p21 family were found to activate cyclin D–CDK4/6 by promoting the association of its component proteins, while inhibiting cyclin E–CDK2 [14,15,16]. Recently, it was shown that only the tyrosine-phosphorylated p21 family activates cyclin D–CDK4/6, and this tyrosine phosphorylation occurs in response to mitogenic signaling [17]. Therefore, p21, induced early after mitogenic stimulation, promotes cell entry into the R-point at the early-/mid-G_1_ phase by activating cyclin D–CDK4/6, but prevents further progression through the R-point by inhibiting cyclin E–CDK2. That is, cells remain at the R-point while *p21* is expressed, but exit from the R-point when *p21* expression is attenuated, indicating that the *p21* gene is involved in R-point regulation. Understanding the molecular mechanisms underlying the mitogen-stimulated immediate early induction of *p21* should enable greater understanding of the mechanisms underlying R-point regulation.

## 3. Role of RUNX3 in *p21* Induction and R-Point Regulation

*RUNX3*, which plays pivotal roles in lineage determination and functions as a tumor suppressor, is frequently inactivated in various types of human cancers, including stomach and lung cancers [18]. Conditional deletion of *Runx3* from mouse lungs results in the development of lung adenomas (ADs), with these pre-invasive lesions progressing to adenocarcinomas (ADCs) following the additional introduction of heterozygous oncogenic *K-Ras* mutations [19]. Deletion of *Runx3* from mouse lung epithelial cells results in the development of lung Ads, and *Runx3^−/−^* mouse embryonic fibroblasts (MEFs) develop into tumors without oncogenic mutations in nude mice [20]. Although these results indicated that cells acquire tumorigenic activity following the deletion of *Runx3*, the mechanism involved remained unclear. An important clue was provided by an analysis of the minimum serum exposure time required for *Runx3*^−/−^ and *Runx3*^+/+^ MEFs to progress to S phase. *Runx3*^−/−^ MEFs required only 1–2 hours, whereas *Runx3*^+/+^ MEFs required at least 4 hours [20]. The short exposure time required by *Runx3*^−/−^ MEFs was similar to that required by *Rb^−/−^* MEFs [21]. Notably, *Runx3* deletion abolished the immediate-early induction of *p21* [20], suggesting that *Runx3* is essential for R-point regulation in MEFs, and that *p21*, a key regulator of the R-point, is a target of *Runx3*. These findings were supported by results showing that the expression of *Runx3* in *Runx3^−/−^* MEFs restored the R-point and the immediate-early induction of *p21,* while abolishing cell tumorigenicity [20]. In addition, the R122C mutation (substitution of arginine 122 to cysteine) in RUNX3, which was identified from human gastric cancer [22], disrupts the R-point [23]. Thus, the tumorigenicity acquired by *Runx3* deletion was associated with a deregulation of the R-point.

## 4. Mechanism for the Induction of R-Point-Associated Genes at the R-Point

Cell commitment at the R-point involves the regulation of several hundred R-point-associated genes, which are induced by exposure to serum for 1–2 h and subsequently suppressed [20]. The *p14-ARF* (hereafter *ARF*) and *p21* genes are included among the R-point-associated genes [19,20,24], but their mechanism of induction early after mitogenic stimulation was originally undetermined. The induction of expression of silent genes requires the target sites within their regulatory regions to be bound de novo by transcription factors. Transcription factors that bind to condensed chromatin independently of other factors, modulate chromatin accessibility, and regulate gene transcription are known as pioneer factors [25,26,27]. To mediate these activities, pioneer factors require a complex network of other proteins, including coactivators, corepressors, histone-modifying complexes, chromatin-remodeling complexes, mediator complexes, and the basal transcription machinery. For example, proteins of the Trithorax group (TrxG) modify histones to activate transcription. TrxG proteins can be classified into two categories: histone modifiers [28] and nucleosome remodelers [29]. TrxG histone modifiers include members of the mixed-lineage leukemia (MLL) family, which methylate H3 at lysine 4 (H3K4-me3, -me2, and -me1), enhancing transcriptional activation. In contrast, TrxG nucleosome remodelers include the SWI–SNF complex, which facilitates binding of transcription factors and the basal transcription machinery. Mediator complexes transduce signals from the transcription activators bound to enhancer regions in the transcription machinery, which is assembled at promoters, as the preinitiation complex, to control the initiation of transcription [30]. Members of the bromodomain (BRD) family of proteins (BRDs) are components of the mediator complex. BRDs are integral to transcription through their interactions with mediator coactivator complexes, which are required for the transcription of various genes [31,32]. BRDs possess two bromodomains, BD1 and BD2, which interact with acetylated histones and acetylated transcription factors.

Runx3 plays a key role in recruiting these chromatin modulators to activate signal-dependent R-point-associated gene expression at the correct target loci at the right time. Immediately after mitogenic stimulation, Runx3 binds to its target chromatin loci and recruits the pRB–E2F complex and p300 acetyltransferase [20]. The interactions are promoted by RAS-activated MEK1 [24]. Runx3 and histones are acetylated by p300 acetyltransferase, with BRD2 subsequently interacting with acetylated Runx3 through its BD1 domain, and with acetylated histone H4 through its BD2 domain [24]. Therefore, RUNX3 guides the pRB–E2F complex and p300 to target loci, with BRD2 binding both acetylated RUNX3 and acetylated histones through its two bromodomains prior to the R-point (Figure 2A).

Subsequently, the C-terminal region of BRD2 interacts with the SWI/SNF chromatin-remodeling complex, MLLs, which act as activated histone modifiers, and TFIID complexes, representing basal transcription machinery [24]. Thus, at the R-point, Runx3 forms a large complex, called the R-point-associated RUNX3-containing activator complex (Rpa-RX3-AC), at target chromatin loci (Figure 2B). This Rpa-RX3-AC complex subsequently opens the chromatin structures of the target loci by replacing the inhibitory histone H3K27-me3 with the activating histone H3K4-me3. Runx3 is an enhancer binding protein, and TFIID is a promoter-binding complex. Therefore, the enhancer interacts with the promoter through the Rpa-RX3-AC complex during the R-point at the target loci, inducing the expression of R-point-associated genes [24] (Figure 2B). This Rpa-RX3-AC complex is maintained, while the RAS-MEK pathway is activated [24]. The activities of RUNX3, including its association with condensed chromatin, its modulation of chromatin accessibility, and its activation of gene expression, fulfill the characteristics of a pioneer factor, making RUNX3 a pioneer factor of the R-point.

## 5. Mechanism for the Suppression of R-Point-Associated Genes after the R-Point

Hypo-phosphorylated pRB is a component of the Rpa-RX3-AC complex that forms 1–2 h after mitogenic stimulation and contributes to R-point commitment [20]. Soon after, CDK4 is recruited to the target locus by interacting with RUNX3. p21, which is induced by the Rpa-RX3-AC complex, facilitates CDK4-–cyclin D1 interactions. Thus, pRB and E2F1, along with CDK4, cyclin D1, and p21, which play key roles in cell cycle regulation, are recruited to the R-point-associated target loci [24] (Figure 2C). At these loci, pRB is phosphorylated at Ser-795 by cyclin D1–CDK4/6. Subsequently, after the R-point, 4 h after mitogenic stimulation, the pRB–E2F1 complex is released from the Rpa-RX3-AC complex, and the expression of R-point-associated target genes is suppressed [24] (Figure 2C). Activation of CDK4 by the cyclin D1–CDK4 interaction triggers the suppression of previously activated R-point-associated target genes, including *p21* and *ARF*.

Proteins in the polycomb group (PcG) modify histones to suppress gene transcription. There are two kinds of PcG complexes, called polycomb repressor complexes 1 and 2 (PRC1 and PRC2). Both complexes consist of multiple proteins, with PRC1 containing BMI1 and ring finger protein 1 (RING1) or 2 (RNF2) [33], and PRC2 containing EED and an enhancer of zeste homologs (EZH1 and EZH2), which trimethylate H3 at lysine 27 (H3K27-me3), a characteristic of inactive chromatin [34]. Cyclin D1, induced soon after mitogenic stimulation, forms a complex with HDAC4 and PRC1 [24]. Therefore, when cyclin D1 binds to CDK4, HDAC4 and PRC1 are also recruited to the Rpa-RX3-AC complex. Because p300-mediated RUNX3 acetylation and histone acetylation are effectively abolished by HDAC4 [35], HDAC4 may play a key role in the deacetylation of RUNX3 and histones, causing the release of BRD2 and other BRD2-associated proteins. Inactivation of chromatin is associated with HDAC-mediated histone deacetylation and H2A ubiquitination at Lys-119, mediated by RNF2, a component of PRC1 [33]. Consistently, H4K12 acetylation was reduced, and H2A-K119-Ub was enriched at the *p21* and *ARF* loci 4–8 h after stimulation [24]. These results demonstrate that cyclin D1, HDAC4, and PRC1 bind to the Rpa-RX3-AC complex through interactions with CDK4. These interactions are facilitated by Rpa-RX3-AC-induced p21, which contributes to the inactivation of chromatin at target loci by deacetylating H4K12 and ubiquitinating H2A (Figure 2C).

At 4–8 h after mitogen stimulation, RUNX3 and BRD2 existed in separate complexes: RUNX3 formed a complex with Cyclin D1, HDAC4, and PRC2, which remained bound to target chromatin loci, whereas BRD2 formed the BRD2–PRC1 complex, which was released from the loci [24] (Figure 2C). Moreover, H3K27-me3 was enriched at these loci. EZH2 is a component of PRC2 that mediates the modification of the inhibitory histone H3K27-me3, suggesting that PRC2, associated with RUNX3, may play a key role in the inactivation of chromatin loci. Because the RUNX3–Cyclin D1–HDAC4–PRC2 complex inactivates chromatin, the complex was named as the R-point–associated RUNX3-containing repressor complex (Rpa-RX3-RE) (Figure 2C).

## 6. Minimally Sufficient Conditions for the Development of Lung Cancer

Many studies have reported that *K-RAS* mutations, the genetic alterations most frequently detected in various cancers, are an early event responsible for the development of lung ADs [36,37,38]. By contrast, the ARF-p53 pathway has been found to effectively defend cells against aberrant oncogene activation [39,40,41], with *p53* mutations being a hallmark of cancer and a prevalent feature of human cancers [42]. Therefore, the development of *K-RAS*-activated cancer might be accompanied by the inactivation of the ARF-p53 pathway. These findings, however, are contradicted by results in human cancers. Evaluation of the key genetic and epigenetic alterations that are responsible for clonal expansion following each step of colon tumorigenesis has shown that colon ADs are initiated by the inactivation of Adenomatous polyposis coli (APC) [43,44] (Figure 3). Moreover, *K-Ras* is activated after AD development, with the loss of *p53* occurring at an even later stage. Although the p53 pathway can defend against colon ADCs, it remains unclear as to whether this pathway can defend against *K-Ras*-activated colon ADs, and, if so, whether the p53 pathway can defend against high-grade, but not low-grade, cancers.

These questions have been partly answered in lung cancer. The progression of lung ADCs from adenomatous growth to carcinomas was found to be similar to the multistep tumorigenesis pathway in colon cancer (Figure 3). Although the activation of *K-Ras* and inactivation of *p53* are frequently detected in lung ADCs, the order of these molecular events has not been clearly established in human lung cancer patients. Rather, the relationship between *K-Ras* activation and *p53* inactivation was analyzed by restoring *p53* expression in *K-Ras*-activated and *p53*-inactivated mouse lung tumors. *p53* restoration eliminated only *Kras*-activated lung ADCs, leaving lung ADs intact in these mice models [45,46]. These results suggested that *p53* is inactivated during late-stage AD or early-stage ADC, later than *K-Ras* activation; this is similar to findings in colon cancer (Figure 3). Previous studies have speculated that this was due to inherent limits in the capacity of the Arf-p53 pathway to respond to a persistent low level of oncogenic K-Ras activity [45,46,47].

However, another possibility remained, that the failure of eliminating lung AD by p53 restoration may be due to disruption of a hidden molecular mechanism responsible for sensing the aberrant persistence of oncogenic signals. The initial step of colon AD development is the inactivation of *APC*, not the activation of *K-Ras*. Similarly, *RUNX3* is frequently inactivated in human atypical adenomatous hyperplasia (AAH) and bronchioloalveolar carcinoma (BAC), which correspond to mouse lung ADs, and inactivation of *Runx3* induces lung ADs in mice [19]. A lone, heterozygous oncogenic *K-Ras* mutation in a large number of cells can also lead to the development of lung ADs, although only a very small number of these cells in a specific cellular context are transformed by oncogenic *K-Ras* [48], indicating that certain spontaneously occurring rare molecular events are involved in the development of *K-Ras*-activated lung cancer. These rare molecular events may occur in only a small percentage of *K-Ras*-activated cells. Thus, the likelihood of these hidden molecular events can be reduced by reducing the number of *K-Ras*-activated cells. Indeed, these same *K-Ras* mutations, with or without *p53* inactivation, in an extremely small number of cells, failed to induce any pathologic lesions for up to 1 year [49]. In contrast, when *Runx3* was inactivated, and *K-Ras* was activated by the same targeting method, lung ADCs and other tumors were rapidly induced and caused lethality in all the targeted mice within 3 months [49]. Therefore, under physiological conditions, in which oncogenic mutations are very rare, *K-Ras* activation alone is not sufficient, whereas its combination with *Runx3* inactivation is sufficient, for lung cancer development. In addition, evaluation of a urethane-induced mouse lung tumor model that recapitulates the features of *K-RAS*-driven human lung tumors showed that *Runx3* was inactivated in both ADs and ADCs, whereas *K-Ras* was activated only in ADCs [49]. Mutations in *p53* were an even later event than *K-Ras* activation [49]. Therefore, the order of the molecular events for the development of mouse lung AD/ADC was likely *Runx3* inactivation → activation of *K-Ras* → loss of *p53* (Figure 3). 

The universal process of malignant transformation involves both genetic damage and oncogenic signaling. These two stresses are signaled to p53 through different pathways. Based on this, p53 plays two important roles in cells: “defense against genome instability”, which consists of sensing and reacting to DNA damage through ATM/ATR kinases, and “defense against oncogene activation”, which consists of responding to oncogenic signaling through the p53-stabilizing protein ARF [50] (Figure 4A). Recent genetic evidence in mice indicates that the ARF-dependent activation of p53 is critical for early-stage p53-mediated tumor suppression. In contrast, ATM/ATR-dependent activation of p53 protects late-stage tumors [50]. Therefore, p53 mutations at relatively late stages of colon and lung tumorigenesis may be associated with the disruption of ATM/ATR-dependent p53 activation (Figure 4B).

Nevertheless, K-RAS-activated AD cells have been found to proliferate in the presence of wild-type ARF and p53. Because heterozygous oncogenic *K-Ras* mutations alone in a small number of cells did not induce lung AD [49], the process of AD development may require the inactivation of the ARF-p53 pathway. The mechanism underlying the inactivation of the ARF-p53 pathway in ADs was unclear. However, *Runx3* is inactivated in most *K-Ras*-activated mouse and human lung ADs [19], with *Runx3* inactivation abrogating the R-point program, which plays a key role in *ARF* induction in response to oncogenic *K-RAS* [24]. Thus, *Runx3* inactivation may inactivate the ARF-p53 pathway in lung ADs, thus providing a mechanism underlying the proliferation of *K-Ras*-activated lung AD cells in the absence of mutated p53 (Figure 4B).

## 7. Mechanism by Which Cells Distinguish Oncogenic from Normal RAS and Defend against Tumorigenesis

Mitogenic signaling activates the GTPase activity of RAS, which decreases to the basal level soon after the signal is transduced to downstream kinase pathways. Oncogenic RAS is a constitutively active form, with GTPase activity not being downregulated. Therefore, heterozygous *RAS* mutations yield cells with 50% of the maximum level of RAS activity [19] (Figure 5A). The ability of cells to sense the duration of 50% rather than maximal RAS GTPase activity may confer protection against oncogenic *RAS*-induced abnormal proliferation. The ability of cells to recognize the aberrant persistence of RAS activity, however, was unclear. For example, oncogenic *K-Ras* expressed at the endogenous level did not activate the ARF-p53 pathway in mouse lungs [20,45,46]. Based on these results, it had been considered that the ARF-p53 pathway does not respond to the aberrant persistence of RAS activity, although the pathway responds to only abnormally high levels of RAS activity [45,46,47]. Mammals, however, were later found to have evolved an effective defense mechanism against a persistent low level of RAS activity [19,24]. When K-RAS is activated by normal mitogenic stimulation, RUNX3 forms Rpa-RX3-AC complexes in a MAPK activity-dependent manner; these complexes transiently induce *ARF*, which in turn transiently stabilizes p53. Soon after the mitogenic surge, MAPK activity is reduced, converting Rpa-RX3-AC complexes to Rpa-RX3-RE complexes and repressing *ARF* expression (Figure 5B). Mitogen-stimulated transient activation of the ARF–p53 pathway does not affect the cell cycle because it occurs only 1–3 hours after mitogenic simulation, and is then silenced before the G_1_/S checkpoint. In contrast, when K-RAS is constitutively activated, the Rpa-RX3-AC complex is maintained, and the expression of *ARF* and *p53* is maintained until the G_1_/S checkpoint, leading to cell death (Figure 5B). These results indicate that cells can effectively defend against an endogenous level of RAS activity, and that the Rpa-RX3-AC complex functions as a sensor and as a decision maker regarding the abnormal persistence of RAS activity [24].

H460 human lung cancer cells were used to determine whether the Rpa-RX3-AC complex-driven activation of the ARF-p53 pathway was sufficient to defend against oncogenic *K-RAS*-induced lung tumorigenesis. In these cells, *K-RAS* was heterozyously mutated but not amplified, and *RUNX3* was inactivated by hyper-methylation. Despite these cells having wild-type *ARF* and *p53*, Rpa-RX3-AC complexes were not formed, and the ARF-p53 pathway was not activated. In contrast, exogenous expression of *RUNX3* led to the formation of Rpa-RX3-AC complexes, which activated *ARF* expression and stabilized p53, thereby inducing cell apoptosis [24]. Expression of mutant *RUNX3*, which is unable to form Rpa-RX3-AC complexes, failed to activate *ARF* expression [24]. Therefore, failure of ARF-p53 pathway activation in H460 cells was due, not to the absence of a mechanism for sensing low endogenous levels of oncogenic K-Ras activity, but to the disruption of the R-point by *RUNX3* inactivation. These findings indicate that cells can recognize the aberrant persistence of RAS activity through the R-point program and kill these cells by activating the ARF-p53 pathway.

## 8. Importance of *RUNX3* in Lung Tumorigenesis

Although several important regulators of cell differentiation govern lung development, deregulation of the differentiation program was generally insufficient to induce AD. *Runx3* is inactivated in nearly all the human and mouse lung ADs and, to date, *Runx3* is the only gene whose inactivation has been reported to induce lung AD [19]. Cancer development is considered to be a biological process that resembles Darwinian evolution: random mutations create genetic variability in a cell population, and the force of selection favors the outgrowth of individual mutant cells that happen to be endowed with advantageous traits. Based on a combination of Darwinian theory and the concept of multistep tumor progression, tumorigenesis is now understood as a succession of clonal expansions [43,44]. Great numbers of cells are required to select cells endowed with advantageous traits. The random inactivation of *Runx3* in normal cells results in a deregulation of the differentiation program [51] and disruption of the R-point program [24]. Deregulation of the differentiation program is likely responsible for the development of AD, although it is not sufficient, whereas disruption of the R-point program likely results in the abrogation of the ARF-p53 pathway-mediated oncogene surveillance mechanism, enabling the subsequent selection of *K-RAS*-mutated cells. Although *K-RAS*-induced lung cancer development can proceed via multiple pathways, the high frequency of *RUNX3* inactivation in *K-RAS*-induced mouse and human lung ADCs suggests that a major pathway involves R-point disruption by *RUNX3* inactivation prior to *K-RAS* activation.

To date, *RUNX3* is the only gene whose inactivation has been shown to be sufficient for both the induction of AD and abrogation of the R-point. These steps may result from multiple molecular events (i.e., one involving each pathway) or a single molecular event, such as *RUNX3* inactivation. Obviously, the probability of deregulation is much higher for events involving a single gene than multiple genes, explaining the importance of *RUNX3* in lung tumorigenesis.

## 9. Tumor Suppressor Genes vs. Oncogenes

Tumor suppressor genes are defined as genes that “help control cell growth,” indicating that tumor suppressors act broadly to inhibit diverse aspects of both normal and neoplastic physiology. By contrast, oncogenes are genes activated by mutations or overexpression of genes that act dominantly to induce tumorigenesis. These terms, however, can overlap, as some proteins with various functions affecting a spectrum of cellular outcomes can enhance and/or suppress tumor pathogenesis. Although *RUNX3* generally acts as a tumor suppressor, *RUNX3* expression can be enhanced during the course of progression of some cancers, with this gene playing a tumor-promoting or oncogenic role. For example, the acquired expression of *RUNX3* in head and neck carcinoma correlates with poor histologic differentiation, invasion, and metastasis [52]. High *RUNX3* expression has also been observed in ovarian cancer, basal cell carcinoma, and skin cancers [18]. Moreover, *Runx3* has been shown to inhibit the early-stage growth of pancreatic cancers but facilitates their metastatic progression at early-stage [53].

The ability of *RUNX3* to exhibit both tumor-suppressing and tumor-promoting activities has been associated with the R-point, a decision-making program for cell proliferation, differentiation, senescence, and apoptosis. The R-point could be deregulated by either the abnormally high expression or inactivation of *RUNX3*. For example, the tumor suppressors *p21* and *ARF* are induced at the R-point and then subsequently suppressed, with *RUNX3* playing key roles in both programs (Figure 2B,C). If *RUNX3* is inactivated, *p21* and *ARF* are not induced, even when an oncogene is activated due to the failure of Rpa-RX3-AC complex formation. In this context, *RUNX3* functions as a tumor suppressor. If, however, *RUNX3* is overexpressed, and Rpa-RX3-AC complex formation is not possible, then RUNX3 may preferentially form Rpa-RX3-RE complexes, suppressing the expression of *p21* and *ARF*. Under these conditions, *RUNX3* functions as an oncogene.

Although *RUNX3* is the only gene to date that has been shown to act as a pioneer factor of the R-point, many pioneer factors of the R-point are likely present in various types of cells. For example, RUNX1 and RUNX2, which are master regulators of hematopoiesis and osteogenesis, respectively [18,54], are involved in R-point regulation [55]. Other master regulators might also play roles in R-point regulation, since development is a sequential process with decisions made at the R-point. Many key R-point regulators may also exhibit ambipotent and context-dependent effects on tumorigenesis. Therefore, we propose designating RUNX3 and similar acting proteins as “decision makers,” with their activities as tumor suppressors or oncogenes being dependent on the intactness of the decision-making machinery in cells.

## 10. Summary

A tumor is defined as an abnormal mass of tissue that forms when cells divide more than they should or do not die when they should. The determination of whether a cell divides or dies is made at the R-point. In theory, cells that make a correct decision at the R-point and correctly execute this decision cannot develop into tumors. Deregulation of the R-point decision-making machinery is involved in the formation of most, if not all, types of tumors [3], suggesting the importance of the R-point in tumor development. Understanding the molecular mechanisms that underlie R-point commitment should provide important insights into how normal cells become tumorigenic. This review summarizes the method by which the R-point distinguishes normal from oncogenic RAS and determines pathways for cell survival or death.

Several fundamental questions underlying cancer development remain to be resolved, including the mechanisms underlying tumor initiation and its rapid recurrence after treatment with anticancer drugs. If oncogene activation is solely responsible for tumor development, then inhibition of the activated oncogene would be able to cure that cancer without the likelihood of recurrence. Although targeted agents that inhibit activated oncogenes have yielded clinical responses, almost all of these malignancies recur. For example, gefitinib was found to effectively eliminate *EGFR*-mutated lung ADCs at the beginning of therapy, but the cancers recurred in 90% of patients within 2 years [56]. Moreover, although oncogenic K-RAS-specific inhibitors have been developed [57], some cancer cells bypass the effects of these inhibitors and resume proliferation [58]. In addition, the knockdown of oncogenic K-Ras in a mouse lung cancer model was found to result in rapid tumor recurrence, not because the gene knockdown was unsuccessful, but because other oncogenes were activated [59]. Because tumor frequency is much lower in normal than in tumor-regressed mice, the rapid activation of secondary oncogenes in the latter suggests that a defense mechanism can be abrogated in established tumors. However, efforts to restore *p53* expression in *K-Ras*-activated mouse lung cancers eliminated only malignant ADCs and failed to eliminate ADs [45,46]. Therefore, it is of great therapeutic importance to understand as to why cancers recur, despite the effective inhibition of the activated oncogene.

Recurrence of lung cancer is due primarily to persistent early lesions that are resistant to oncogene inhibitors. These early lesions do not contain activated oncogenes. Therefore, to eradicate cancers, it is necessary to understand their mechanisms of initiation. Inactivation of *RUNX3* is thought to be responsible for the initiation of lung ADs, as well as for abrogating the R-point-regulating ARF-p53 pathway. The proliferation of *K-RAS*-activated lung AD cells with wild-type ARF and p53 results from *RUNX3* inactivation, which abrogates the ARF-p53 pathway in lung ADs.

Normal cells recognize the aberrant persistence of oncogenic K-RAS signals through their RUNX3-containing R-point-associated activator (Rp-RX3-AC) complexes, which sense the duration of RAS signals and regulate the ARF-p53 pathway. p53 deletions may be required at later stages of AD for abrogation of the ATM/ATR-p53 pathway. K-Ras-activated mouse lung ADs acquire secondary oncogene activation rapidly, because R-point associated oncogene surveillance mechanisms are abrogated in the ADs. *RUNX3* restoration has been shown to eliminate *K-RAS*-activated tumors in a human lung cancer cell line. It would be exciting indeed if *Runx3* restoration eliminates *K-Ras*-activated lung cancer in an animal model. If that turns out to be the case, *RUNX3* will be a promising target for curative cancer therapy.

## Figures and Tables

**Figure 1 cells-12-00708-f001:**
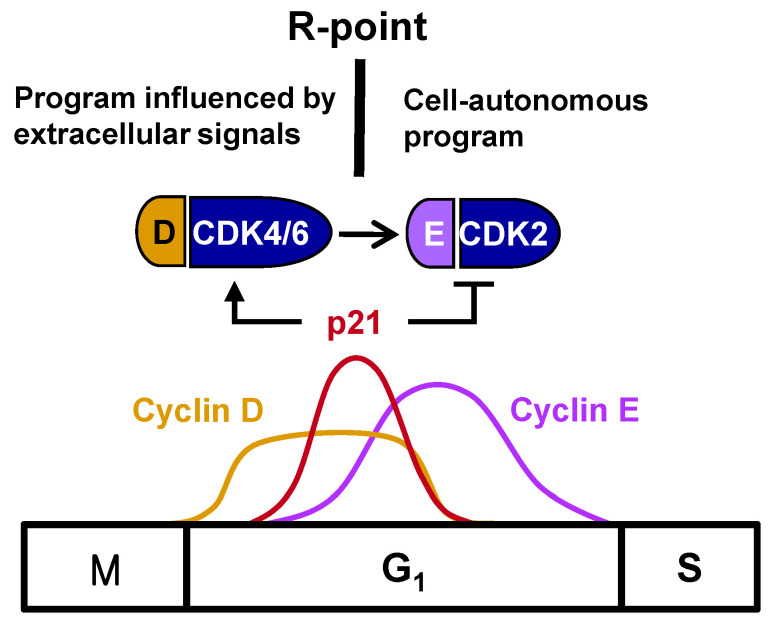
Fluctuation of levels of cyclins D and E and p21 through the cell cycle. The levels of cyclins D and E and p21 fluctuate markedly as cells progress through the cell cycle. The CDK inhibitors of the p21 family stimulate the formation of the cyclin D–CDK4/6 complex while inhibiting the formation of other cyclin–CDK complexes, including cyclin E–CDK2. Extracellular signals strongly influence the levels of D-type cyclins during the early G_1_ phase. However, the levels of the other cyclins, including cyclin E, are controlled by intracellular signals and precisely coordinated with cell cycle progression. The cyclin E–CDK2 complex is activated after cells pass through the R-point, followed by the formation of the remaining cyclin–CDK complexes through a cell-autonomous program.

**Figure 2 cells-12-00708-f002:**
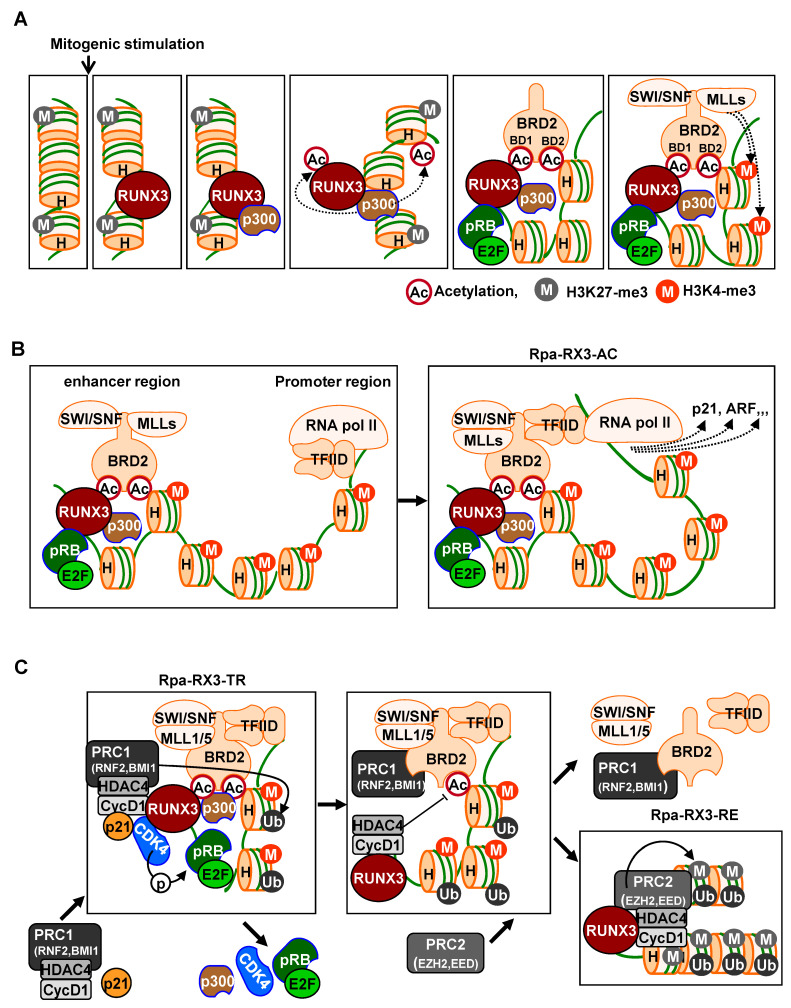
Sequential molecular events for R-point decisions. (**A**) In response to mitogenic stimulation, RUNX3 opens target loci. Upon mitogenic stimulation, RUNX3 binds to the enhancer regions of target loci within inactive chromatin, as indicated by H3K27-me3. pRB-E2F1 and p300 associate with RUNX3. p300 acetylates RUNX3 and histones. BRD2 binds to acetylated RUNX3 through its first bromodomain (BD1) and to H4K12-ac through its second bromodomain (BD2). Subsequently, SWI/SNF and MLL1/5 bind to the C-terminal region of BRD2. At this time point, inhibitory histones (H3K27-me3) are eliminated, and activating histones (H3K4-me3) are enriched at these loci. (**B**) RUNX3 forms an R-point-associated RUNX3-containing Activator (Rpa-RX3-AC) complex at the R-point. While RUNX3 binds to the enhancer region and recruits its coactivator (p300), histone-modifying enzymes (MLLs), and chromatin-remodeling complex (SWI/SNF), the basal transcription machinery (TFIID) is recruited to the promoter region of the target loci. The TFIID binds to the C-terminal region of BRD2 to form Rpa-RX3-AC. Moreover, the enhancer interacts with the promoter through Rpa-RX3-AC during the R-point. (**C**) Rpa-RX3-AC complex is converted to Rpa-RX3-RE after the R-point. Two hours after mitogenic stimulation, CDK4 (associated with p21) binds to RUNX3 and becomes an additional component of Rpa-RX3-AC. At this point, the cyclin D1–PRC1 complex forms separately from the Rpa-RX3-AC complex. Downregulation of the RAS-MEK signal results in the maturation of the cyclin D1–PRC1 complex in the cyclin D1–HDAC4–PRC1 complex, which binds to Rpa-RX3-AC through the interaction between cyclin D1 and CDK4, a component of the Rpa-RX3-AC complex, yielding Rpa-RX3-TR. Activation of CDK4 through its association with cyclin D1 is critical for the inactivation of the chromatin loci and the dissociation of the entire complex. RNF2, a component of the PRC2, contributes to the enrichment of an inactive chromatin marker (H2A-K119-Ub, H2A ubiquitination at Lys-119) at this locus. If the RAS signal is constitutively activated, the cyclin D1–PRC1 complex fails to mature into the cyclin D1–HDAC4-PRC1 complex, and consequently cannot form Rpa-RX3-TR. Therefore, if R-point commitment is normal, cells expressing constitutively active RAS cannot progress through the R-point into the S phase. If the mitogenic signal is downregulated in a normal manner, Rpa-RX3-TR dissociates (4 h after stimulation) into two complexes, the RUNX3–Cyclin D1–HDAC4 and BRD2–PRC1–SWI/SNF–TFIID complexes, which remain associated with chromatin. This is followed by the association of PRC2 with RUNX3–cyclin D1–HDAC4 to form Rpa-RX3-RE, which remains on the chromatin. EZH2, a component of PRC2, contributes to the enrichment of an inactive chromatin marker (H3K27-me3) at this locus.

**Figure 3 cells-12-00708-f003:**
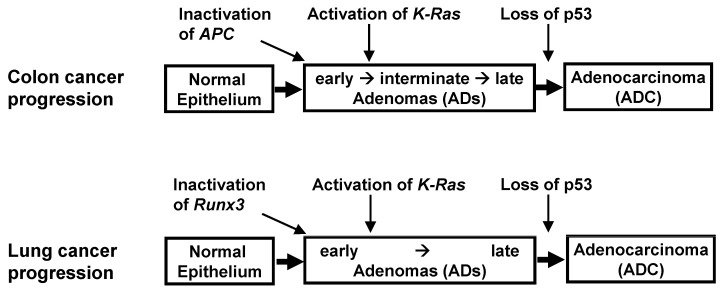
Sequential molecular events occurring during multistep tumor progression. Most colorectal and lung adenocarcinomas develop through a multistep tumorigenesis pathway. Tumors show development from normal tissue, to adenoma (AD), to adenocarcinoma (ADC), and ultimately progress to multiple types of invasive tumors. Molecular events occurring at each step are indicated.

**Figure 4 cells-12-00708-f004:**
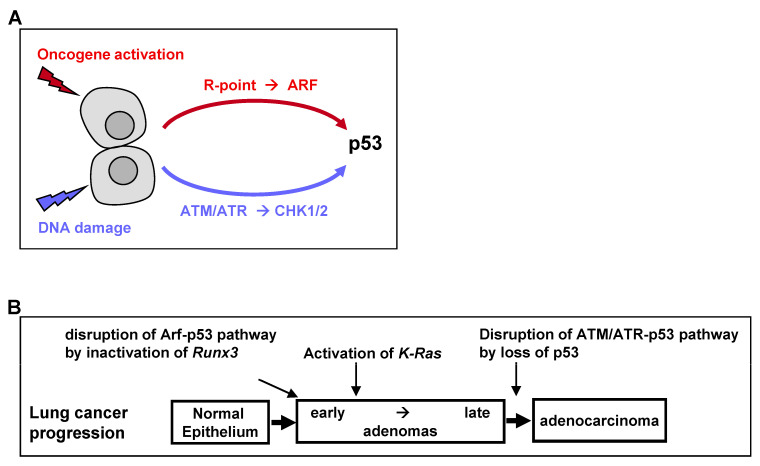
Inactivation of p53 tumor suppressor pathways. (**A**) Two major pathways trigger p53 activation. Aberrant oncogene activation is sensed by the R-point-associated complex, which induces the expression of *ARF*, inactivating HDM2 and stabilizing p53. DNA damage stress is sensed by the ATM/ATR kinases, activating the CHK1/CHK2 kinases, which stabilize p53. (**B**) Inactivation of *p53* tumor suppressor pathways during multistep tumor progression. AD development is characterized by disruption of the Arf-p53 pathway due to the abrogation of the R-point, most frequently by *RU3* inactivation. This may result in the selection of *K-Ras*-activated cells, which acquire a proliferative advantage. At the AD stage, the ATM/ATR → CHK1/2 → p53 pathway is functional. The pathway is disrupted at a late stage of AD by *p53* mutation.

**Figure 5 cells-12-00708-f005:**
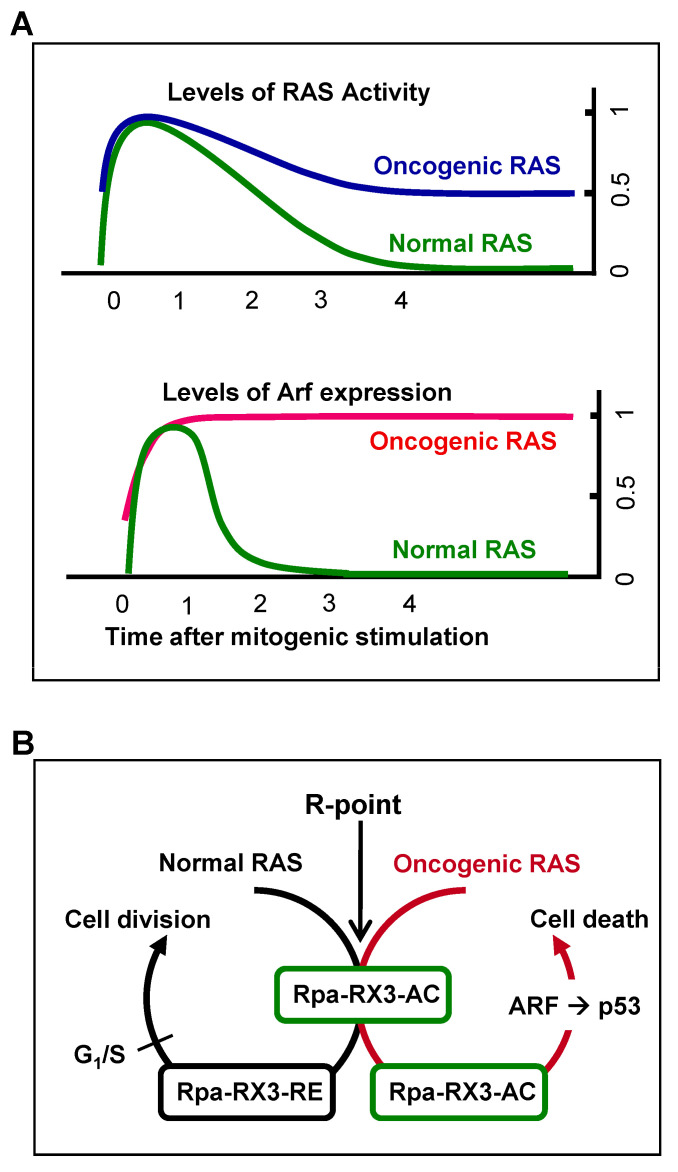
Mechanism for sensing constitutive RAS activation. (**A**) Normal RAS activity is downregulated to the basal level soon after mitogenic stimulation. While RAS is activated, *ARF* is expressed. In normal cells, *ARF* is expressed for only a short time (1–3 h after mitogenic stimulation), followed by its suppression when RAS activity is downregulated. However, heterozygous mutations of *RAS* result in the maintenance of 50% of the maximum level of RAS activity. This persistent RAS activity maintains *ARF* expression until the G_1_/S checkpoint is reached. (**B**) Schematic illustration of the R-point-associated oncogene surveillance mechanism. Formation of the Rpa-RX3-AC complex is triggered by the RAS-MEK pathway 1 h after serum stimulation. The complex binds to the *ARF* promoter through RUNX-binding sites and induces *ARF* expression. After the R-point (4 h after mitogenic stimulation), the RAS-MEK pathway activity is downregulated. Rpa-RX3-AC complexes are converted to Rpa-RX3-RE complexes, which suppress *ARF* expression. However, constitutively activated RAS signaling inhibits the conversion of Rpa-RX3-AC to Rpa-RX3-RE complexes and prolongs *ARF* expression, which drives cells toward apoptosis. These series of molecular events enable cells to distinguish normal mitogenic signals from abnormal oncogenic K-RAS signals, thereby constituting an R-point-associated oncogene surveillance mechanism.

## Data Availability

There is no data to share.

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
