# Peer review of "Role of RUNX3 in Restriction Point Regulation"

_cells, 2023, doi:10.3390/cells12050708_

Round 1

Reviewer 1 Report

The review “Role of RUNX3 in Restriction Point Regulation” reports the role of R-point participates in oncogenic surveillance. Here are the comments for the author.

1.     The abstract should be modified because authors have started the abstract using this line “Throughout the cell cycle, cells monitor their cumulative exposure to specific signals over 10 time and make the critical decision to pass through the restriction (R)-point.”. Here, authors should start with cell cycle and then discuss about the R-point. So that any reader can understand easily.

2.     The sentence “Deregulation of this machinery is markedly associated with tumorigenesis, making the identity of the molecular mechanisms that govern R-point decision making one of the fundamental issues in tumor biology” should be rephrases and splited into two sentences as it is hard to understand in current format.

3.     Are authors sure “RUNX3 is frequently inactivated in tumors by epigenetic alterations.” That only one transcription factor is responsible for that? What about other transcription factors?

4.     Please use the consistent cases throughout the manuscript like here is one example check “Runx3”, “K-Ras” in the sentence “In particular, RUNX3 15 is downregulated in most K-RAS-activated human and mouse lung adenocarcinomas (ADCs). Tar- 16 geted inactivation of Runx3 in mouse lung induces adenomas (ADs) and markedly shortens the 17 latency of ADC formation induced by oncogenic K-Ras.”

5.     Authors have used keywords “ARF”, and “p21” both have not discussed in abstract.

6.     The sentence “In normal cells, reductions in growth factor levels, such as by the removal of serum, prevent the onset of S-phase.” Do not have any reference even it is not appropriate to use word “normal cells” this prevention does with the cell lines also. Here are the link of studies which have shown the addition/ removal of growth factors affect the cell cycle. In addition, serum deprived media and other factors like hydroxyurea/ 2DG arrest the cell cycle at G1 stage. These studies must be discussed here.

https://doi.org/10.1016/j.lfs.2018.09.055

https://doi.org/10.1371/journal.pone.0206364

7.     The statement is misleading “RUNX3, which plays pivotal roles in lineage determination and functions as a tumor 101 suppressor, is frequently inactivated in various types of human cancers, including stomach and lung cancers”. However, PU.1 is a transcription factor that also a tumor suppressor in acute myeloid leukemias. PU.1 have also important role in lineage determination, wherein when interact with GATA-1 it leads to erythropoiesis and when it interacts with c-Jun it leads to myelopoiesis. So, both studies should be discussed important to know it in addition to RUNX3.

https://doi.org/10.1038/sj.onc.1211004

10.2147/IJN.S303235

8.     It seems that Figure 2, and Figure 6a are taken from any other paper, then authors should insert the reference in the figure caption.

9.     Figure 3 is very hard to follow. So, authors should provide the heading at the top of each section and mention clearly what they wanted to convey in the figure. The figure should be self-explanatory.

10.  The heading “Mechanism for the Induction of R-Point-Associated Genes at the R-Point” is confusing, so it should be changed.

Reviewer 2 Report

This is a very well written review about Runx3 and R-point, which is a extremely complex process. Especially, the section 5 nicely draws a conclusion that positions Runx3 as a pioneering factor. Overall, this manuscript will be a valuable asset to the field of Runx3.

1. In line 58, the authors introduced “E2F transcription factor family”, but it does not explain it. Either explaining about E2F transcription factor family genes induced by pRB, or removing it will make it easier to follow.

2. In the section of lines 62-74, the authors bring an interesting point about the complexity of G1/S cell cycle network which is plagued with complicated forward and backward looping. It will be nice for readers if the authors can conclude this paragraph with a few sentences about their opinion on these “seemingly paradoxica results”.

3. In line 73, “dependence” sounds a little weird to this reviewer. What about “dependency”?

4. In lines 77-80. This reviewer feels a little reluctant to accept this description in which the authors suggests that this sequence of events is pushed forward by the previous events just because they are in the linear relationship. Therefore, the uses of the word “driving” might be a little too provocative.

5. In lines 113-114, it is rather not appropriate to claim that the R-point is disturbed in Runx3 -/- MEFs just because both cell lines share a common phenotype (the short exposure time).

6. In line 196, is “H3K27-mer” a spelling error?

7. The section 5 describes the roles of Runx3 as a pioneering factor. One could wonder if other members of Runx proteins (Runx1 or Runx2) have any roles in this scenario. Maybe the authors can describe the expression of Runx proteins in MEFs and their kinetics during the cell cycle progression.

7. In line 217, is the use of the word “categories” appropriate here? Maybe “kinds” might be better.

8. In lines 221-234, HDAC4 is pointed out as a key player. This reviewer wonders if only the HDAC4 play a role in this process since there are several HDACs.

9. In line 242, “name” should be “named”.

10. The section 6 alone is a marvelous piece of writing to describe this interesting process of ADC development, but it might be nice if the authors put a few sentences to connect this section with R-point process at the end of the section. Or merging the section 6 and 7 might suffice.

11. In lines 249-250, this reviewer feels a little reluctant to accept the word “must” here.

12. In lines 254 and 278, what are APCs here?

13. In line 276, it is not clear what is “the failure”.

14. In line 292, “killed” might not be appropriate. How about “caused lethality in”?

15. In line 303, what is “two stresses”?

16. In line 327, “therefore” sounds a little awkward here.

17. Lines 391-393 and lines 402-403 appear to state the opposite thing. Also, the uses of “deregulation” and “dysregulation” are little confusing.

18. Lines 393-394, adding the percentages of AD cases in which RUNX3 inactivation is observed might be informative.

19. In lines 451-460, the authors use “divides or not” and “divides or dies” multiple times. However, this reviewer feels that they have quite different meanings. Should it be more consistent in this section?

20. Considering the sentence of lines 454-456, one might wonder how come RUNX3 is the only known pioneering factor of the R-point. Can the authors elaborate a little on this point such as what kinds of technical limitation exist to further understand this?

21. Bae-group’s own finding about R122C in RUNX3 (doi.org/10.1016/j.jcmgh.2022.01.010) is not mentioned in this review, but it might be informative if it is included, just a personal opinion though.

22. (Arguably of course) all RUNX proteins form complexes with CBFb proteins. Therefore, it might be appropriate to briefly describe about RUNX-CBF association in this review.
